# Long-Term Snow Height Variations in Antarctica from GNSS Interferometric Reflectometry

Elisa Pinat [1,*], Pascale Defraigne [1], Nicolas Bergeot [1,2], Jean-Marie Chevalier [1,2] and Bruno Bertrand [1]

1   Royal Observatory of Belgium, 1180 Brussels, Belgium; pascale.defraigne@oma.be (P.D.); nicolas.bergeot@oma.be (N.B.); jean-marie.chevalier@oma.be (J.-M.C.); bruno.bertrand@oma.be (B.B.)
2   Solar-Terrestrial Center of Excellence, 1180 Brussels, Belgium
*   Correspondence: elisa.pinat@oma.be

**Abstract:** Acquiring reliable estimates of the Antarctic Ice Sheet surface mass balance is essential for trustworthy predictions of its evolution and future contribution to sea level rise. Snow height variations, i.e., the net change of the surface elevation resulting from a combination of surface processes such as snowfall, ablation, and wind redistribution, can provide a unique tool to constrain the uncertainty on mass budget estimations. In this study, GNSS Interferometric Reflectometry (GNSS-IR) is exploited to assess the long-term variations of snow accumulation and ablation processes. Eight antennas belonging to the Polar Earth Observing Network (POLENET) network are considered, together with the ROB1 antenna, deployed in the east part of Antarctica by the Royal Observatory of Belgium. For ROB1, which is located on an ice rise, we highlight an annual variation of snow accumulation in April–May (~30–50 cm) and ablation during spring/summer period. A snow surface elevation velocity of $+0.08 \pm 0.01$ ma$^{-1}$ is observed in the 2013–2016 period, statistically rejecting the "no trend" null hypothesis. As the POLENET stations are all located on moving glaciers, their associated downhill motion must be corrected for using an elevation model. This induces an increased uncertainty on the snow surface elevation change determined from GNSS-IR. Among the eight stations analyzed, only three of them show a long-term snow height variation larger than the uncertainties. One is located on the Flask Galcier in the Antarctic Peninsula, with a decrease of more than 4 m between 2012 and 2014, with an uncertainty of 2.5 m. The second one is located on the Lower Thwaites Glacier where we observe, between 2010 and 2020, a snow surface drop of 10 m, with a conservative uncertainty of 1 m. The third station, located on the West Antarctic Ice Sheet (WAIS) divide, shows on the opposite an upward motion from 2005 to 2019, of 1.2 m with an uncertainty of 0.4 m. The snow surface change of the other POLENET stations analyzed is smaller than the uncertainty associated with the glacier slope.

**Keywords:** GNSS-IR; Antarctica; snow accumulation

## 1. Introduction

Enhancing our knowledge of the Antarctic Ice Sheets (AIS) Mass Balance is crucial to improve predictions of its future evolution and associated changes in global sea level. The AIS hold enough water to rise global sea level by 58 m, and even a small change in the AIS mass balance can produce global-scale noticeable effects [1]. Several methods are used to determine ice sheet mass changes: altimetry, sensitive to volume changes, and gravimetry, sensitive to mass changes and the input–output method, which is sensitive instead to dynamic changes in the ice [2]. In the input–output method, where mass balance is expressed as the difference of the input and output fluxes, the input flux is given by the Surface Mass Balance (SMB). SMB consists in the net balance between the processes of accumulation and ablation on a glacier surface: snowfall, rainfall, snowdrift, snow melt, percolation, and refreezing at the surface. Snow accumulation and ablation play therefore a key role in Antarctic surface mass balance estimations. GNSS (i.e., GPS and GLONASS)

Interferometric Reflectometry (GNSS-IR) is a well-established technique that uses the variations in the signal-to-noise ratio (SNR) measurements to sense the antenna near field environment [3–5]. Reflected signals, usually considered as a detriment in positioning methods, are here exploited and turned into a source of information on the reflecting surface. The frequency of the SNR interference sinusoidal pattern depends indeed on the distance between the phase center of the GNSS receiver antenna and the reflecting surface, and on the signal wavelength. As in [5], the vertical distance between the antenna phase center and the horizontal reflecting snow/air surface is referred to as the *reflector height*. Accounting for the ellipsoidal correction, the antenna vertical position is combined with the reflector height to monitor the snow surface elevation at the antenna site. Applied to antennas in Antarctica, GNSS-IR allows to retrieve snow height variations. GNSS-IR has already been applied on different antennas in Antarctica [6,7], using however dedicated stations for short-term campaigns with dedicated logistics and budget. In this study, we instead present results from nine antennas installed across the continent for completely different purposes, such as, for example, the DRIL GPS station, originally installed to monitor the vertical and lateral movements of the ANDRILL drilling site on the McMurdo Ice Shelf [8], that can provide additional information on the Antarctic SMB with long term snow height variations at their location. The technique is applied to different antenna glaciological environments: the ROB1 antenna was deployed by the Royal Observatory of Belgium in East Antarctica on an ice rise, and the eight GNSS stations from the Polar Earth Observing Network (POLENET) [9] network (DRIL, FLSK, KHLR, LPRD, LTHW, REC1, WAI2, WAIS) are instead situated on ice sheets or ice shelves, covering part of West Antarctica and the Antarctic Peninsula.

## 2. Data and Methods

### 2.1. ROB1 and POLENET Antennas

The geographical location of the antennas used in the analysis is displayed in Figure 1.

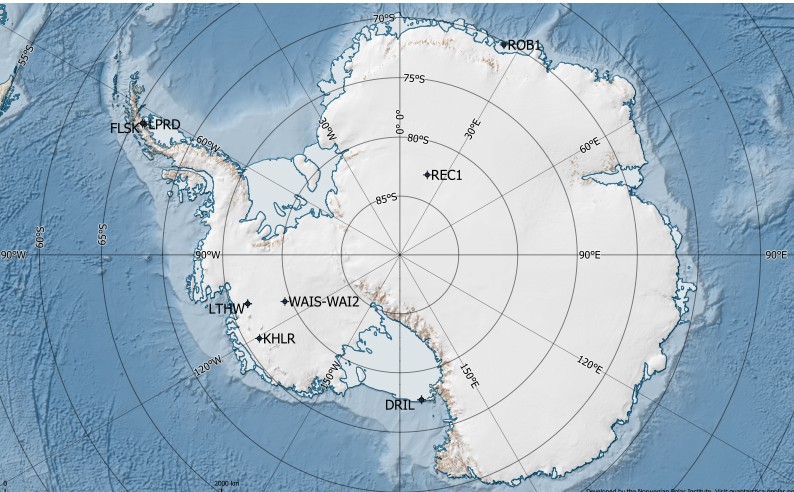

**Figure 1.** Map of Antarctica showing the geographical positions of the ROB1, DRIL, FLSK, LPRD, LTHW, REC1, WAIS/WAI2, and KHLR antennas. Source: Quantarctica, Detailed Basemap Norwegian Polar Institute, 2018.

Deployed by the Royal Observatory of Belgium on the Derwael Ice Rise, in the coastal Queen Maud land, East Antarctica, ROB1 has been originally installed to monitor ice motions [10] and has provided continuous data from late 2012 to early 2016. POLENET is a large international and multidisciplinary project that combines seismic and GNSS instrumentation at remote sites in Antarctica. Antenna deployment started over two decades ago, but major networks of remote stations were first deployed beginning with the International Polar Year 2007–08, counting nowadays roughly hundred stations in total, some of them permanent and some temporary. Among these, eight stations were selected as being

located over glaciers and allowing for GNSS-IR to be optimally applied: DRIL, FLSK [11], KHLR [12], LPRD [13], LTHW [14], REC1 [15], WAIS [16], and WAI2 [17]. As the WAIS and WAI2 antennas are co-located and one followed the other in time, these are treated as a single dataset. Datasets used in this study are publicly available on the UNAVCO website. These antennas have not been deployed for GNSS-IR studies, but, due to their favorable position, the application of this method is an additional scientific outcome. All the data series used in this work present either some data gaps or antenna mast additions, that, however, do not prevent the snow height long term analysis. Unfortunately, most of the log files of the different POLENET antennas do not report when a mast is added and the antenna height is changed with respect to the surroundings. Table 1 summarizes all the time windows covered by the different datasets, together with the latitude and longitude coordinates of the antennas on their initial location at the beginning of the time window.

**Table 1.** Summary of the different antenna dataset time spans used in this analysis and longitude and latitude at the beginning of the period.

|  | Location | Start | Stop | $Lon_{start}$ | $Lat_{start}$ |
|---|---|---|---|---|---|
| ROB1 | Derwael Ice Rise | 05 December 2012 | 14 January 2016 | 26.345 | −70.241 |
| DRIL | Ross Ice Shelf | 14 November 2010 | 12 December 2013 | 171.511 | −77.582 |
| FLSK | Flask Glacier | 06 February 2010 | 12 July 2014 | −62.897 | −65.752 |
| KHLR | Kohler Glacier | 24 January 2010 | 03 September 2019 | −120.729 | −76.155 |
| LPRD | Leppard Glacier | 13 February 2010 | 22 May 2013 | −62.903 | −65.953 |
| LTHW | Lower Thwaites Glacier | 12 December 2009 | 30 August 2020 | −107.782 | −76.458 |
| REC1 | Recovery Glacier | 10 January 2009 | 03 June 2011 | 18.905 | −82.812 |
| WAIS | West Antarctic Ice Sheet Divide | 01 December 2005 | 26 January 2010 | −112.054 | −79.467 |
| WAI2 | West Antarctic Ice Sheet Divide | 08 December 2010 | 19 January 2019 | −112.054 | −79.467 |

*2.2. GNSS Interferometric Reflectometry*

GNSS-IR is a well-established technique that uses SNR variations to sense the antenna near-field environment. With the term SNR we indicate here the C/N0 ratio, expressed in decibel-Hertz (dB-Hz), i.e., the ratio of the carrier power and the noise power per unit bandwidth, and not the signal-to-noise power ratio in a given bandwidth, expressed in decibels. GNSS-IR has already proven its robustness in different applications, as for example measuring surface soil moisture [3], snow depth [4], or firn density [18]. Contrary to snow depth studies, where density information and bare soil height are required to assess the snow water equivalent, snow height studies focus only on the variation of the snow surface elevation over time. The common denominator of GNSS-IR studies is the analysis of SNR patterns created by the interference between the direct and indirect signals at the antenna (see Figure 2). Reflected signals, often known as multipath and normally considered as a detriment in positioning methods, are here exploited and turned into a source of information on the reflecting surface.

Considering only the multipath component, SNR data for a single satellite arch are modeled as [19]

$$SNR(e) = A(e) \sin\left(\frac{4\pi R}{\lambda} \sin e + \phi\right), \tag{1}$$

where $e$ is the satellite elevation angle, $\lambda$ is the signal wavelength, $\phi$ is a phase constant, and R is the reflector height. $A(e)$, representing the amplitude of the SNR data, depends on the transmitted signal power, the antenna gain pattern, and the composition of the surface, but will not be investigated in this study. If R is fixed, surface soil moisture can be derived by tracking changes in the phase term $\phi$ [3], term that is also not considered in this work and fixed to zero. Snow depth or firn densities are instead addressed by studying the variations of the intrinsic multipath frequency $2R/\lambda$. The same quantity $2R/\lambda$ is also the key parameter of the analysis on snow height variations presented in this work. R

represents the distance between the antenna phase center and the reflecting surface, which in our case is the air/snow interface. As the signal penetrates, layers of snow surfaces of different densities will also produce distinct reflections, but the reflection contrast and the interference effect are greatest at the surface [18]. This study has been performed using the L1 GPS wavelength, with $\lambda \sim 19$ cm. GNSS antennas are designed to mitigate multipath as much as possible. This goal is well achieved for elevation angles higher than about $30°$, but, below this value, multipath still exists and produces interference with the direct signal. If the antenna is surrounded by a plane surface, these interferences generate periodic variations in SNR data. As an example, Figure 3a shows the SNR of the G13 satellite rising track recorded by the ROB1 antenna on 5 December 2012. In this study, SNR data are retrieved from the satellite Receiver Independent Exchange Format (RINEX) files using the software ROB-IONO [20], and we consider a valid elevation range from $5°$ to $25°$. The computation of reflector heights is performed using a self-developed code in Python based on that in [19], following the strategy detailed hereafter. SNR data are first linearized using the formula $SNR_{linear} = 10^{\frac{SNR_{dB}}{20}}$. Afterwards, a second order polynomial that describes the direct component resulting from the transmitted signal power an the antenna gain pattern is fitted to the $SNR_{linear}$ data, as shown in Figure 3a for the same satellite track. This direct component is then removed in order to isolate the multipath contribution, reported in Figure 3b, the principal ingredient of the frequency analysis. In fact, the frequency of the interference oscillations is $2R/\lambda$, parameter that is assumed constant for all the rising and setting tracks of all satellites. By determining the dominant frequency, it is therefore possible to retrieve the reflector height R.

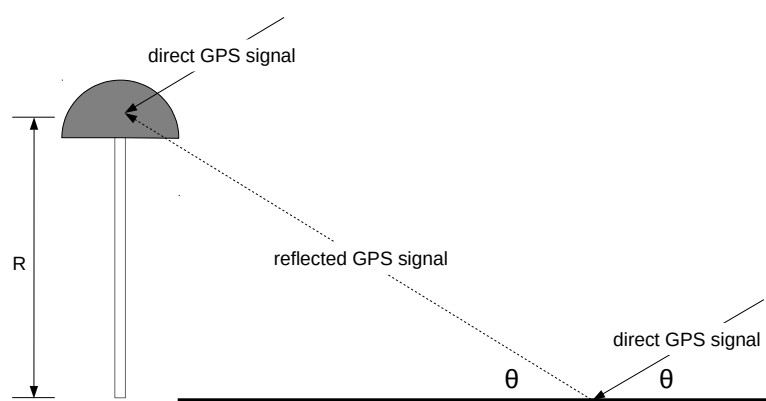

**Figure 2.** At low elevation angles, the interference between reflected and direct signals at the antenna enables GNSS Interferometric Reflectometry (GNSS-IR), a technique to study the reflecting surface properties, among which the variation of the *reflector height* R over time.

The Lomb–Scargle periodogram of the multipath contribution to SNR variations allows to obtain the dominant frequency in the oscillating data. Figure 3c shows the Lomb–Scargle periodogram of the multipath component of the same satellite track as before. In the x axis, frequencies are represented as reflector heights. We can observe a peak at a height of 1.7 m, which is therefore considered as the dominant reflector height for this track.

Satellite tracks are considered valid if they contain more than 55 signal-to-noise ratio values, corresponding to a 27.5 min rising or setting track with a 30 s sampling rate. The value was adapted from in [18] to better match the geometry of the GPS constellation in polar regions. The Lomb–Scargle periodogram requires two input parameters defined by the user, i.e., the desired precision and the maximum height, that defines the frequency window where to search for the peak. The maximum reflector height considered in this analysis has been empirically determined and is set to 7 m, while the precision is set to 0.05 m. Additional quality cuts are applied to improve the method performance: the periodogram result is considered as valid only if the main peak is greater than four times the mean power of the periodogram in the window 3–7 m, and only if no secondary peak is obtained with a magnitude larger than one third of the main one. Results that pass the

quality cuts are then summed, to obtain the reflector height corresponding to the maximum of the sum of all the valid periodograms within one day. This method differs from taking the daily mean of all the satellite results also because it is not straightforward to establish the uncertainty associated with the result. Peak widths and Gaussian error bars should generally be avoided when reporting uncertainties in the context of a periodogram analysis [21]. A false alarm probability approach is instead preferred, as it aims at understanding the significance of a peak by measuring the probability that a dataset with no signal would lead to a peak of a similar magnitude owing to coincidental alignment among the random errors. False alarm probability studies have not been performed until this point and will not be presented in this analysis. Following the work in [18], reflector height uncertainties considered in this work are at the cm level.

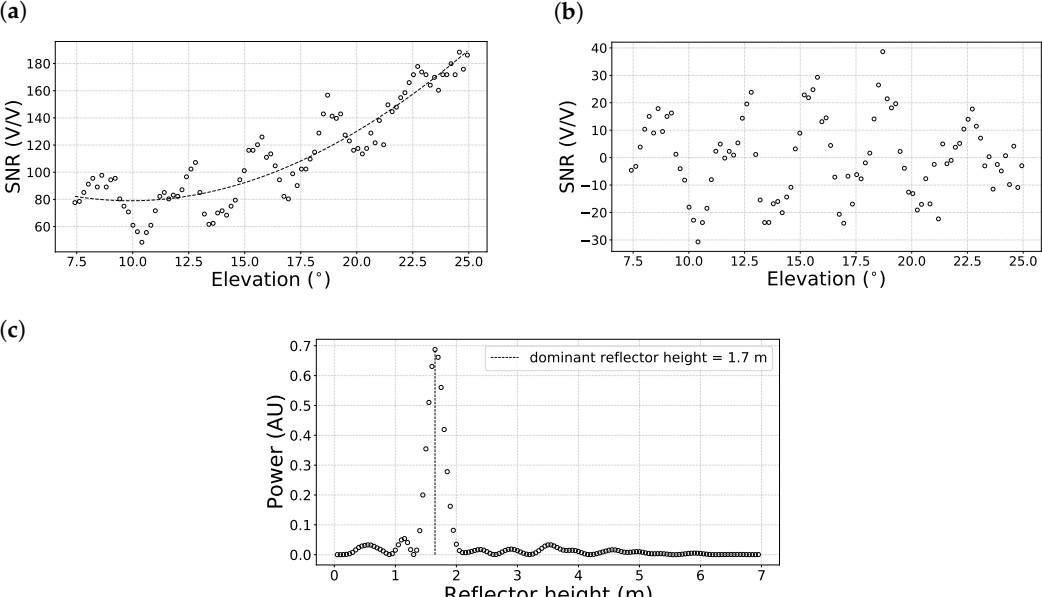

**Figure 3.** (**a**) Linearized signal-to-noise ratio (SNR) data for G13 satellite rising track on December 5, 2012 at ROB1 station. The interference between the direct and reflected component is evident for elevation angles lower than 30°. The gray trend, driven by the transmitted signal power and the antenna gain pattern, is represented by a second order polynomial and removed in the multipath analysis. (**b**) Isolated multipath contribution after the removal of the second order polynomial describing the signal direct component. (**c**) Lomb–Scargle periodogram of the same satellite track. Frequencies on the x axis are reported as reflector heights.

For each satellite track, the full sensing footprint of any technique relying on reflected GNSS signals can be approximated by the first Fresnel zone [22], the ellipsoidal area that defines the region where most of the reflected signal energy is transmitted to the antenna. The size of this zone depends on the satellite elevation angle *e*, the carrier wavelength $\lambda$ and the distance between the antenna and the reflector, the reflector height R, which is vertical for an horizontal reflector. The semi-minor and semi-major axes of the first Fresnel zone are given by

$$b = \sqrt{\frac{\lambda R}{\sin e} + \left(\frac{\lambda}{2\sin(e)}\right)^2} \tag{2}$$

$$a = \frac{b}{\sin(e)} \tag{3}$$

and the orientation of the major axis is defined by the direction of the satellite-antenna vector. For an antenna located two meters above the snow, the semi-major axis ranges from roughly 24 to roughly 2 m for an elevation of 5 or 25 degrees respectively. The semi-minor axis instead ranges from roughly two to one meter.

### 2.3. Measuring the Snow Surface Elevation

Alone, reflector height variations calculated with the GNSS-IR technique do not suffice to evaluate snow height contributions at the antenna site. One additional essential parameter is the variation of the antenna elevation with respect to the underlying bedrock. Indeed, the antenna mast sinks into the snow, reducing the reflector height independently from any snow accumulation. The GNSS Precise Point Positioning (PPP) Atomium software [23] is used in this study to calculate the horizontal and vertical antenna displacements over time. The contribution of the antenna sinking is then given by the decrease of the antenna position vertical component, with respect to its position on the first day of data. For some stations, this contribution is so important that a mast needs to be added almost yearly to prevent the antenna loss in the snow. The impact of this addition however reflects both on the reflector height measurement $R(t)$ and on the vertical position $Z_{\mathrm{geo}}(t)$, canceling out in the final measurement $E(t)$. $E(t)$ represents the ellipsoidal elevation of the snow surface at a fixed Eulerian point whose coordinates coincide with the initial location of the GNSS antenna, and is therefore called throughout the text the Eulerian snow surface elevation [18]. It is expressed as

$$E(t) = Z_{\mathrm{geo}}(t) - R(t), \tag{4}$$

where $Z_{\mathrm{geo}}(t)$ is the elevation calculated by Atomium and $R(t)$ is the reflector height value obtained with GNSS-IR. As the downward motion of the antenna due to ice dynamics is taken into account in the $Z_{\mathrm{geo}}(t)$, resulting variations in $E(t)$ represent therefore the snow-only contribution, as illustrated in Figure 4.

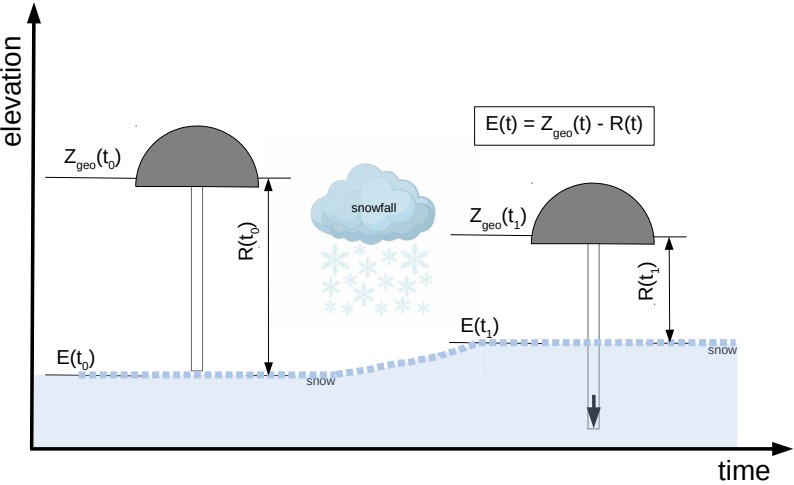

**Figure 4.** Antenna schematics in areas with predominant vertical motion. The antenna horizontal position is considered fixed in time, and its elevation variations are only due to its vertical sinking combined with potential snowfall. $E(t)$ indicates the reflecting surface elevation, $R(t)$ is the reflector height, and $Z_{geo}(t)$ is the vertical position of the antenna phase center

Written as it is, Equation 4 is however only valid if the antenna is located on an ice rise, where the primary antenna motion occurs in the vertical direction. Elsewhere, another important parameter to be considered is the possible antenna downhill drift due to its location on a moving glacier. In this case, the vertical motion of the antenna phase center Zgeo(t) is not only due to the mast sinking, but a second component associated to the glacier motion must be considered (see Figure 5).

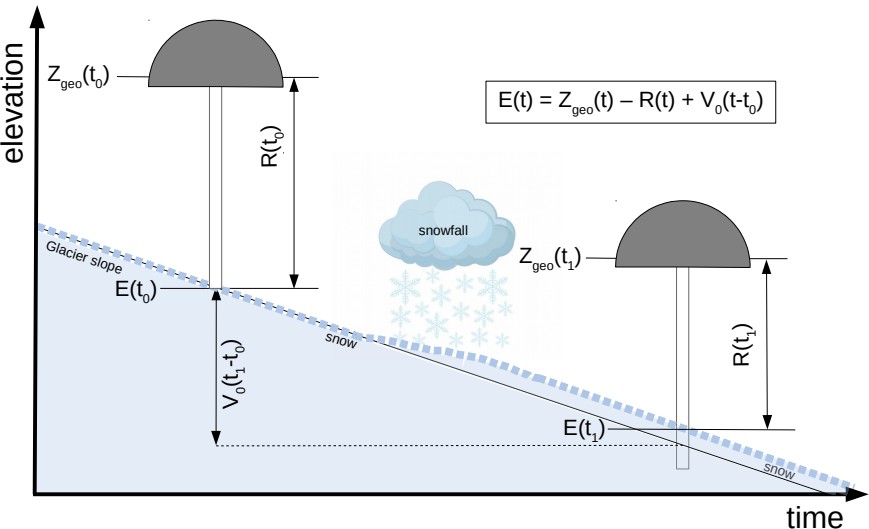

**Figure 5.** Schematics of an antenna located on a glacier. In this case, as time passes, an additional component enters the elevation calculation, which is the downhill motion due to the physical slope of the glacier.

Therefore, to calculate the Eulerian surface elevation fixed with the antenna initial position, we need to ideally remove the descent rate $dH/dt$ due to the glacier slope, called $V_0$. The easiest assumption for this extra term is to consider a steady state vertical drop: $E_{\mathrm{downhill}}(t) = -V_0(t - t_0)$, and to consider an altitude model of Antarctica to estimate its contribution. The variation in the Eulerian elevation assumes therefore the following form:

$$E(t) = Z_{\mathrm{geo}}(t) - \mathrm{R}(t) + V_0(t - t_0) \tag{5}$$

representing the GNSS-based snow elevation varying with time at a fixed Eulerian point coincident with the initial position of the antenna. We use the horizontal motion computed by Atomium to determine the initial and final geographical coordinates of the antenna, and extract from an Antarctica altitude model the altitude difference between both locations. The model considered in this study is the Reference Elevation Model Antarctica REMA, which provides elevations with a spatial resolution of eight meters [24]. This is a first order approximation, considering a linear elevation profile between the starting and end point. For each elevation value, the model provides also a $1\sigma$ associated uncertainty. When combining these to estimate the final uncertainty, we adopt a conservative approach, considering uncertainties as independent.

As the time span considered increases, the $V_0(t - t_0)$ approximation for the downhill flow becomes less accurate. Additionally, as the Eulerian point is fixed while we are actually dealing with a moving object, the resulting snow accumulation estimations assume a uniform snow distribution between the initial location and the final one (maximum one kilometer in the stations considered in this study), reducing therefore the impact of local phenomena that might affect snow height measurements.

## 3. Results

GNSS-IR allows to retrieve the daily reflector height, representing the distance between the reflecting surface and the antenna phase center. Therefore, higher values of the reflector height indicate that the antenna is farther away from the reflecting snow surface, while lower values indicate an antenna closer to the surface. Daily reflector heights and snow height variations as a function of time are presented in this section.

### 3.1. Antennas on an Ice Rise: ROB1

As previously mentioned, the station ROB1 was deployed in 2013 on the Derwael Ice Rise, in the coastal Queen Maud land, East Antarctica, for a period of 3 years. Due to the subsidence of the antenna associated with ice dynamics, a mast of 1.6 m was added to the setup in December 2014 and 2015. The daily values of the reflector height computed with the GNSS-IR approach described in Subsection 2.3 are displayed in Figure 6. For each year, we see a slow decrease of the reflector height, followed by a sudden jump corresponding to the mast addition. From these results it is, however, not possible to distinguish the contributions from the antenna subsidence due to ice dynamics and from the snowfall. This requires taking into account the positions of the GNSS antenna computed from GNSS measurements. Before that, note that GNSS-IR poorly resolves reflector heights below twice the signal wavelength used, in this case about 40 cm [25]. From in situ measurements we know, however, that the ROB1 antenna sunk below this working threshold, and reflector height results of about 50 cm, immediately before the mast additions, are slightly overestimated. In situ measurements of the antenna height are reported on Figure 6.

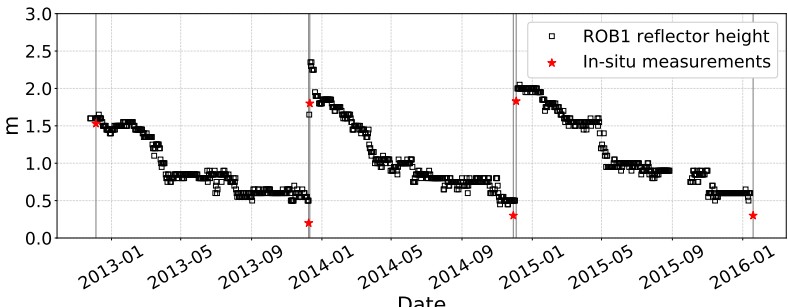

**Figure 6.** GNSS-IR daily reflector height for the ROB1 antenna. In situ measurements of the antenna height are reported as red stars.

Due to its location on an ice rise, we know that ROB1 should move predominantly in the vertical direction. When considering ROB1 position over time as computed by PPP, displayed in Figure 7, we indeed find yearly mean velocities of $v_N = 0.16$ ma$^{-1}$, $v_E = -0.19$ ma$^{-1}$, and $v_U = -1.36$ ma$^{-1}$. A constant conservative uncertainty of 2 cm has been considered for the daily positions given by PPP solutions, leading to $2\sigma$ uncertainties on the estimated velocities smaller than 0.005 ma$^{-1}$ in all components.

The Eulerian snow surface elevation $E(t)$ is calculated using Equation 4, i.e., combining the reflector height and the antenna vertical motion. The corresponding results are reported in Figure 8. Annual cycles are clearly visible, with snow accumulation mainly in April–May (30–50 cm) followed by ablation during austral spring/summer time. The Eulerian surface elevation trend at the ROB1 station is assessed by performing a linear regression over the entire period, regression that finds a slope of +0.08 ± 0.01 ma$^{-1}$. ROB1 results are summarized in Table 2.

To assess the statistical significance of this result, a hypothesis test is performed on the dataset, considering as null hypothesis the "no trend" hypothesis. The statistical test used is the python implementation of the Mann–Kendall, used to analyze time series data for consistently increasing or decreasing monotonic trends. Among the multiple result of the test, we are interested in the determined trend ("no trend", "increasing", or "decreasing"), and in the p-value, that determines the statistical significance of the result with respect to the null hypothesis. For ROB1, the obtained trend is "increasing", with a zero p-value indicating the incompatibility of the dataset with the null hypothesis.

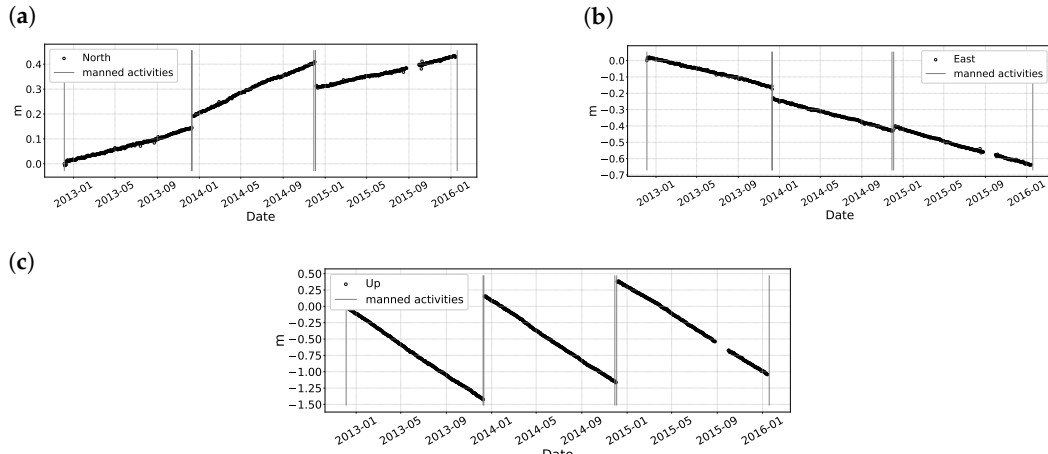

**Figure 7.** ROB1 antenna displacement in the north (**a**), east (**b**), and up (**c**) directions with respect to the first day position. Gray vertical solid lines indicate manned activity dates, where a mast of 1.6 m was added to prevent the antenna sinking in the snow.

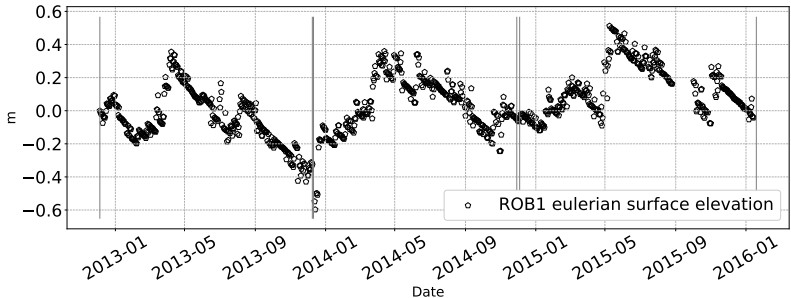

**Figure 8.** Eulerian surface elevation at the ROB1 antenna location over the 05-12-2012 to 14-01-2016 period.

**Table 2.** ROB1 results summary. Non-reported $2\sigma$ uncertainties are $<0.005$ ma$^{-1}$ in all north, east, and up components.

| Ellipsoidal H$_{\text{start}}$ | $v_{\text{N}}$ (ma$^{-1}$) | $v_{\text{E}}$ (ma$^{-1}$) | $v_{\text{U}}$ (ma$^{-1}$) | REMA $\Delta h$ (m) | Annual Snow Variations | $v_{E(t)}$ (ma$^{-1}$) |
|---|---|---|---|---|---|---|
| 453.4 | 0.16 | $-0.19$ | $-1.36$ | - | $\pm 30/50$ cm | $0.08 \pm 0.01$ |

*3.2. Antennas on a Moving Glacier: POLENET Antennas*

Time series of daily reflector heights for the eight POLENET antennas computed following our GNSS-IR approach are reported in Figure 9.

The only available in situ measurement is for the WAI2 antenna is reported in Figure 9g, and corresponds to an antenna height of 2.40 m, measured on 8 December 2011. This validates our results of reflector height presented in Figure 9g.

Also in this case, likewise ROB1, the reflector height parameter is not the final quantity we are interested in, as our final goal are the snow height variations around the antenna site (Figure 10).

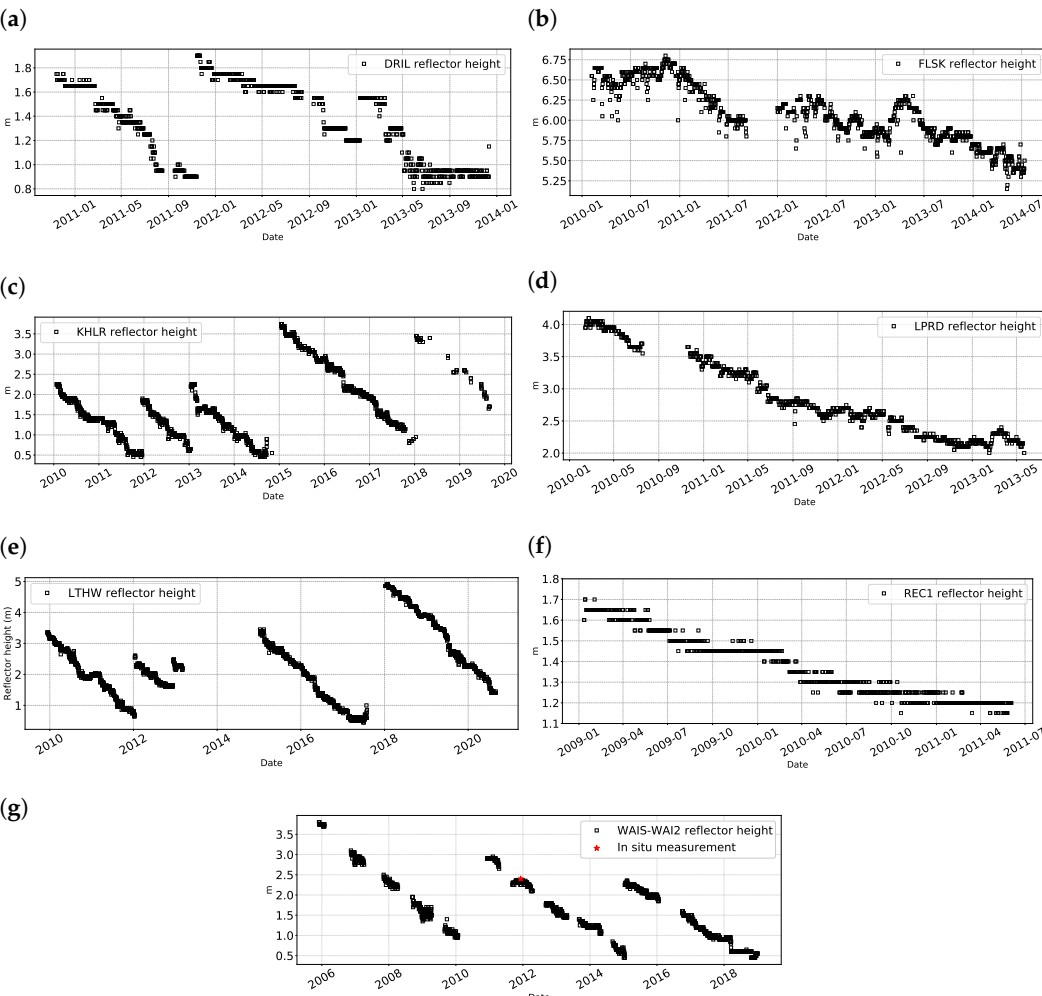

**Figure 9.** Reflector height time series for the eight POLENET (Polar Earth Observing Network) antennas considered in this analysis: (**a**) DRIL, (**b**) FLSK, (**c**) KHLR, (**d**) LPRD, (**e**) LTHW, (**f**) REC1, and (**g**) WAIS-WAI2.

For the POLENET stations, significant horizontal displacements are observed in the station positions computed with GNSS PPP, as shown in Table 3. Snow height variations, i.e., the Eulerian surface elevation at the initial position of the antenna, are therefore computed using Equation (5), with the $V_0$ term calculated from the REMA elevation difference values reported in Table 3. The uncertainties on the REMA $\Delta h$ result from the propagation of the uncertainties provided by the REMA elevation model at each location. In Table 3, we report also the total snow variation over the entire time window and the estimated linear velocity of the Eulerian surface elevation: depending on the antenna Eulerian surface elevation time series, this parameter is either calculated over the entire data time window or piecewise, handling separately parts with different trends. The reported values are the maximum and minimum velocities retrieved for the antenna, with the corresponding year reported in parenthesis. For the POLENET antennas, the major component of the uncertainties on the Eulerian surface elevation velocities comes from the uncertainty on the REMA $\Delta h$. Contrary to ROB1, POLENET antennas do not show clear annual cycles. Among the eight stations analyzed, only three of them show a long-term snow height variation larger than the uncertainties. The FLSK antenna, located on the Flask Galcier in the Antarctic Peninsula, shows a decrease in its Eulerian surface elevation of more than 4 m between 2012 and 2014, with a conservative uncertainty of 2.5 m. For the second antenna, LTHW located on the Lower Thwaites Glacier, we observe a snow surface drop of ten meters in ten years, between 2010 and 2020, with a conservative uncertainty

of one meter. Finally, WAIS-WAI2, located on the WAIS divide, shows on the opposite an upward motion from 2005 to 2019, of 1.2 m with an uncertainty of 0.4 m. The snow surface change of the other POLENET stations analyzed is smaller than the uncertainty associated with their glacier slope.

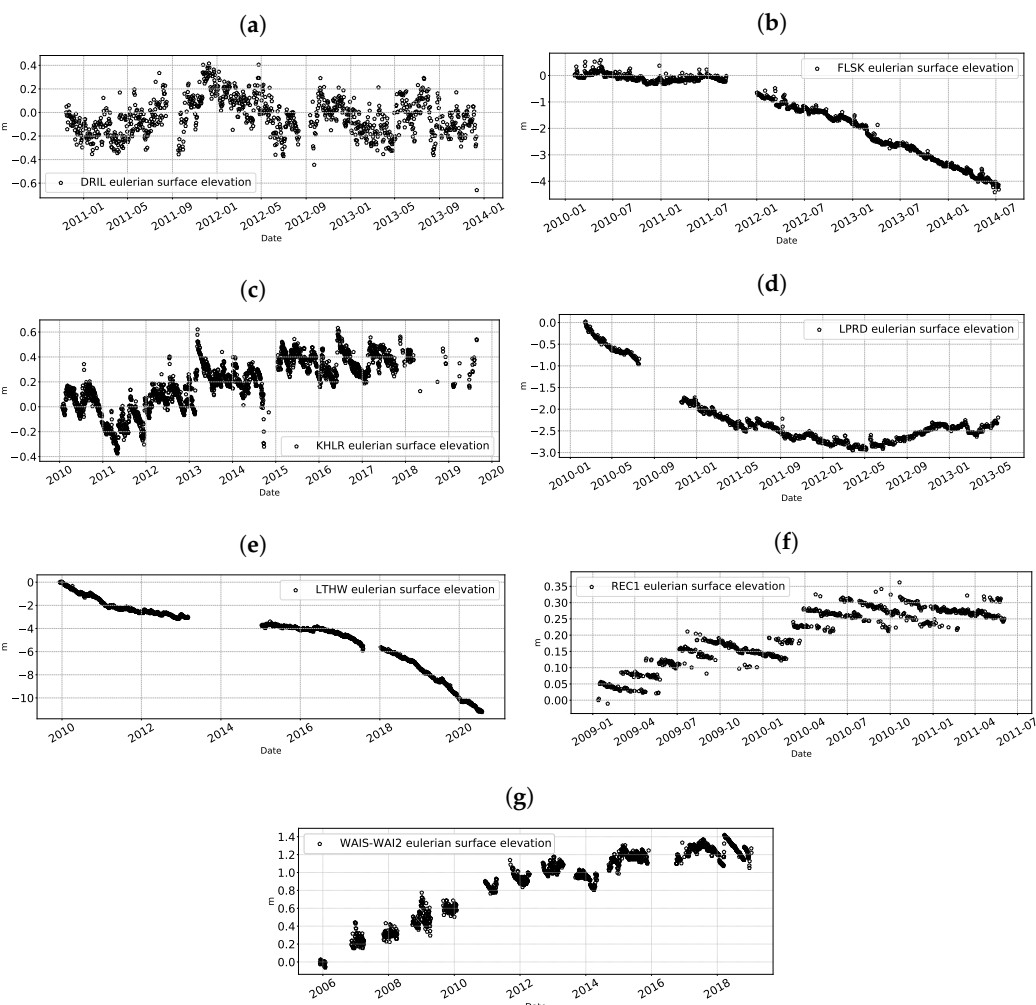

**Figure 10.** Eulerian surface mean elevation for the eight POLENET antennas considered in this analysis: (**a**) DRIL, (**b**) FLSK, (**c**) KHLR, (**d**) LPRD, (**e**) LTHW, (**f**) REC1, and (**g**) WAIS-WAI2.

In 2012, six antennas were active: DRIL, FLSK, KHLR, LPRD, LTHW, and WAIS-WAI2. In Figure 11, we display the 2012 yearly maximum velocity of the Eulerian surface elevation with down going red arrows and up going green arrows. For comparison, the 2019 Eulerian surface elevation velocity for LTHW is reported on the left with an orange arrow. LTHW is the only station for which we have recent data, as KHLR presents too few points in 2019 to perform a linear regression.

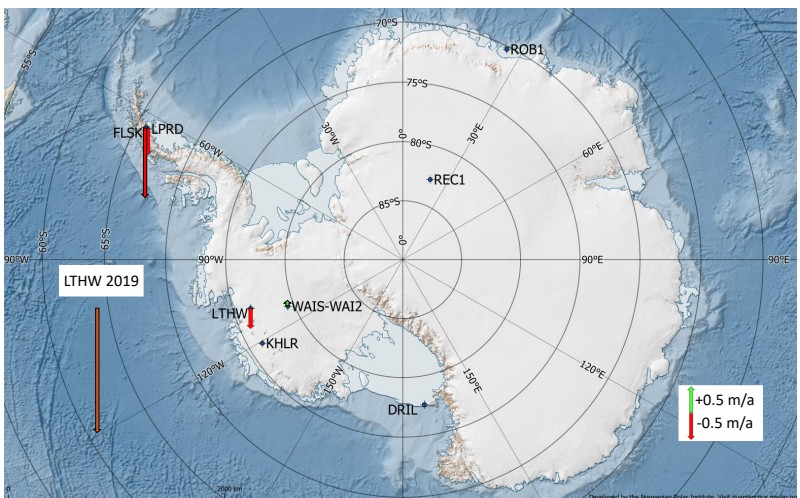

**Figure 11.** 2012 yearly maximum velocity of the Eulerian surface elevation. For comparison, LTHW Eulerian surface velocity for 2019 is reported on the left with an orange arrow.

**Table 3.** Summary of antennas results.

| Antenna | Dates | $\Delta_N$ (m) | $\Delta_E$ | REMA $\Delta h$ (m) | $\Delta_{snow}$ (m) | $v_{minE(t)}$ (ma$^{-1}$) | $v_{maxE(t)}$ (ma$^{-1}$) |
|---|---|---|---|---|---|---|---|
| ROB1 | Start: 05 December 2012 Stop: 14 January 2016 | −0.6 | 0.4 | - | - | 0.08 ± 0.01 | 0.08 ± 0.01 |
| DRIL | Start: 14 November 2010 Stop: 12 December 2013 | 2359.3 | −243 | 0.6 ± 0.5 | - | 0.00 ± 0.16 | 0.00 ± 0.16 |
| FLSK | Start: 06 February 2010 Stop: 12 December 2013 | 194.4 | 1058.4 | 23.1 ± 2.5 | −4.5 | −0.16 ± 0.56 (2010) | −1.40 ± 0.56 (2012) |
| KHLR | Start: 24 January 2010 Stop: 26 August 2019 | 88.0 | 85.1 | 2.5 ± 0.6 | +0.4 | 0.06 ± 0.06 | 0.06 ± 0.06 |
| LPRD | Start: 13 February 2010 Stop: 22 May 2013 | 64.8 | 940.5 | 13.4 ± 2.6 | −2.5 | 0.48 ± 0.79 (2013) | −1.74 ± 0.79 (2010) |
| LTHW | Start: 12 December 2009 Stop: 31 July 2020 | 3596.4 | 666.4 | 14.4 ± 1.0 | −11.2 | −0.37 ± 0.09 (2012) | −2.48 ± 0.09 (2019) |
| REC1 | Start: 10 January 2009 Stop: 03 June 2011 | 4.4 | − 28.2 | 0.0 ± 0.4 | +0.3 | 0.11 ± 0.17 | 0.11 ± 0.17 |
| WAIS-WAI2 | Start: 01 December 2005 Stop: 19 January 2019 | −16.1 | −16.7 | 0.0 ± 0.4 | +1.2 | 0.15 ± 0.17 (2014) | 0.15 ± 0.17 (2010) |

## 4. Discussion

A direct comparison with existing independent Surface Elevation Changes datasets (SECs) of Antarctica is challenging, as our results are at specific locations and over given time windows that do not always match those in the literature. One exception is the LTHW antenna, located on the extensively studied glacier Thwaites. Surface elevation changes of the Amundsen Sea sector, where the Thwaites Glacier is located, are shown in Figure 3 of [26], which displays the Cryosat-2 radar altimeter elevation change data between 2010 and 2013 with overlaid multiple airborne laser measurements between 2009 and 2012. From this picture, we observe a drop rate of the surface elevation of the same order of magnitude as what found in this study with GNSS-IR, i.e., between 0 and −1 ma$^{-1}$. Both Cryosat and airborne laser highlight negative elevation changes, with an average bias between the two as high as 0.5 ma$^{-1}$. GNSS-IR, providing a completely independent measurement of the surface elevation change at specific locations, could help identify the origin of this bias. The uncertainty on the GNSS-IR results is indeed small (see Table 3) compared with the 0.16 ma$^{-1}$ uncertainties on the Cryosat and airborne results. Furthermore, we can see from Figure 10e that the drop of the LTHW snow surface elevation is accelerating with time, reaching a maximum rate of −2.5 ma$^{-1}$ in the time window 2019–2020, while the negative trend is significantly softer before that time. The Thwaites Glacier is indeed known to be melting at an accelerated pace, as evidenced from the accelerated retreat of its grounding zone [27,28].

A more global Antarctic dataset of surface elevation changes was released in 2019 [29]. Time series of surface elevation changes have been built based on 25 years of satellite radar

altimeter observations and a regional climate model. The online graphical user interface allows to easily access information on different drainage basins, for which the surface elevation change trends $dh/dt$ ma$^{-1}$ are provided as mean values over specific periods. In Table 4 we report a comparison of the basin mean value (on the left) with our antenna results (on the right).

**Table 4.** Comparison between the new Antarctic 25 year elevation trends on the left and the results of this study on the right.

| Basin | Period | $dh/dt$ (ma$^{-1}$) | Antenna(s) | Period | $dh/dt$ (ma$^{-1}$) |
|---|---|---|---|---|---|
| 6 | 2012–2016 | 0.032 | ROB1 | 01/2013–01/2016 | 0.076 |
| 26 | 2012–2016 | −0.181 | FLSK | 01/2012–07/2014 | −1.404 |
| | | | LPRD | 05/2012–05/2013 | 0.481 |
| 19 | 2012–2016 | −0.023 | WAIS-WAI2 | 2012–2016 | 0.066 |
| 21 | 2012–2016 | −0.415 | LTHW | 2012–2016 | −0.374 |
| 20 | 2012–2016 | −0.207 | KHLR | 2012–2016 | 0.064 |

The main observation is that, except for the just discussed LTHW station on the Thwaites Glacier, there are substantial differences between the local GNSS-IR measurements and the average basin surface elevation change. Significant differences are even found between nearby stations located in the same basin, as is the case for LPRD and FLSK. These differences are well above the uncertainties on our numerical velocity estimations. This clearly reflects the local character of the snow surface elevation changes with time. These local measurements are however a useful input to constrain both regional climate models and satellite data. They are currently particularly scarce, both in terms of spatial and time coverage. Therefore, the installation and maintenance of new GNSS stations on the snow in Antarctica should be encouraged. This could not only help an in-depth regional study of precipitation rates, as suggested in [30], but also enable the generation of surface elevation changes maps as in [31]. Moreover, if we focus especially in the Antarctic Peninsula where the POLENET antenna network is dense, measurement of snow surface elevation changes using GNSS-IR technique could be useful in resolving some observational anomalies like the discrepancy between the small average change in ice sheet mass compared to the expected fluctuations in snow accumulation highlighted in [26].

## 5. Conclusions

In this paper, the GNSS Interferometric Reflectometry technique has been applied to Antarctic stations to retrieve long data series of snow accumulation/ablation in different regions and on different glaciological environments. One station, ROB1, is located on the Derwael ice rise in East Antarctica and moves roughly only in the vertical direction. This feature allows us to access detailed information on snow height variations and seasonal cycles at the specific antenna location. A positive snow surface elevation trend of $+0.08 \pm 0.01$ ma$^{-1}$ was observed in this location in the period 2013–2016. The other GNSS stations presented in this paper are located on glaciers and move horizontally tens or even hundreds of meters per year. This additional factor requires a correction term for the glacier motion, which introduces as well an additional uncertainty on the long-term variations of the snow surface elevation.

The REMA elevation model has been used to model the glacier slope, with conservative uncertainties on the slope ranging from 0.4 to 2.6 m depending on the site location. The results can be summarized as follows.

- For the LPRD station, located on the Leppard Glacier in the Antarctic Peninsula, we observe an overall decrease of the snow level of about three meters from 2010 to 2012, followed by a more stable period. However, the order of magnitude of the decrease is the same as the uncertainties of the REMA elevation model.
- For the FLSK antenna, located on the Flask Glacier in the Antarctic Peninsula, we observe a global decrease of more than four meters during the period 2010–2013, with an uncertainty of 2.5 m from the REMA elevation model.

- A prominent meltdown is observed for the LTHW antenna, located in the lower part of the Thwaites Glacier, with a global decrease between 2010 and 2020 of more than ten meters of the snow surface, a decrease one order of magnitude higher than the uncertainty obtained from the REMA elevation model.
- A positive trend in the snow surface elevation of 1.2 m from 2005 to 2019 (with an uncertainty of 0.4 m) was observed for station WAIS-WAI2, located on the West Antarctic Ice Sheet ice flow divide.

The other antennas of the POLENET network—KHLR located on the Kohler Glacier in West Antarctica, and REC1, located on the Recovery Glacier—display slightly positive trends, which however are of the same order of magnitude as the uncertainty in the REMA elevation model. The DRIL antenna, located on the Ross Ice Shelf, does not show a variation in the snow elevation.

Moreover, we compared our results with external state-of-the-art estimations of regional snow elevation changes over time. These estimates, based on satellite data, airbone laser data, or regional climate models, showed similarities but also discrepancies up to $1 \, \text{ma}^{-1}$ with our results, depending on the location of the comparison. These differences can be explained by local effects, which are not visible in regional models.

Finally, our study demonstrates that GNSS antennas vertical time series together with GNSS-IR can generate long-term continuous time series of snow surface elevation potentially at every suitable location of already deployed GNSS antennas, without requiring additional field costs. Extending the available stations would serve for multiple purposes: on one side they could be used to constrain for example firn density models, as presented in [18]; on the other hand, they could be used as independent measurements to validate regional models, and play a role in the calibration and bias resolution between different techniques. Our study is a first step in that direction.

**Author Contributions:** Conceptualization, N.B., P.D. and E.P.; methodology, E.P., P.D. and N.B.; software, E.P.; validation, E.P., P.D. and N.B.; formal analysis, E.P., P.D. and N.B.; investigation, E.P., P.D. and N.B.; resources, N.B.; data curation, N.B., J.-M.C. and E.P.; writing—original draft preparation, E.P.; writing—review and editing, E.P., P.D., N.B., J.-M.C. and B.B.; visualization, E.P., P.D. and N.B.; supervision, P.D. and N.B.; project administration, P.D.; funding acquisition, P.D., N.B. and B.B. All authors have read and agreed to the published version of the manuscript.

**Funding:** This research received no external funding.

**Institutional Review Board Statement:** Not applicable.

**Informed Consent Statement:** Not applicable.

**Data Availability Statement:** UNAVCO data for the POLENET stations are publicly available. ROB1 data are available upon request.

**Acknowledgments:** Authors would like to acknowledge the geospatial support and DEMs for this work provided by the Polar Geospatial Center under NSF-OPP awards 1043681, 1559691, and 1542736; the POLENET Consortium and the Belgian Science Policy Office project ICECON.

**Conflicts of Interest:** The authors declare no conflict of interest.

## Abbreviations

The following abbreviations are used in this manuscript:

| | |
|---|---|
| GNSS | Global Navigation Satellite System |
| GNSS-IR | GNSS Interferometric Reflectometry |
| GNSS PPP | GNSS Precise Point Positioning |
| REMA | Reference Elevation Model of Antarctica |
| SEC | Surface Elevation Changes |
| SMB | Surface Mass Balance |
| SNR | Signal-to-noise Ratio |
| WAIS | West Antarctic Ice Sheet |

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
