# Peer review of "Long-Term Snow Height Variations in Antarctica from GNSS Interferometric Reflectometry"

_remotesensing, doi:10.3390/rs13061164_

Round 1

Reviewer 1 Report

This paper uses GNSS-Interferometric-Reflectometry to monitor the long-term snow height in Antarctica. The topic is important for developing new sensors to monitor the snow height in remote areas. Overall, the analysis is detailed, and the data is relatively reliable. It is also nice the authors considered most of the real conditions in Antarctica. However, this article has not been well summarized and organized. There are many technical details but no reliable analysis on the "Long-Term Snow Height Variations" indicated in the title, even without the significance testing for snow height change.

Here are some minor comments:

  1. Line 4-7, this manuscript's topic, as the title indicates, is the snow height, but not the snowfall. I suggest the authors directly indicate the importance of snow height by about two sentences, while there are too many sentences to approach the topic (line 1- 7).
  2. Line 12-16, any significant testing?
  3. Line 44. what are the 'purposes'?it should be explicitly presented here.
  4. Figure 1. It is not a standard geographic map without the coordination, the compass, and other necessary elements.
  5. Line 158-159. This resolution may be too rough to determine the vertical change due to the antenna drift.
  6. Line 194-197. It is better to show these in-situ measurements in the figures.
  7. Figure 7. Change the symbol for the 'manned activity'. It is too similar to the reference lines.
  8. Line 227-229. all in-situ observations should be plotted in the figure for comparison.
  9. Line 212-229. This paragraph gives many details about the availability of measurements, with a little chaos, while the readers look forward to the summarized results. It is better to table all these details. And, these details may be better to be introduced in DATA part.
  10. In the RESULTS part, I see many details about how the data are available and how the elevation changes were determined, but not the well-organized / direct results.
  11. PART of Conclusions and perspectives. What are your results implications? Here is only a simple summarization but without the conclusions or perspectives.

Author Response

Dear Reviewer, 

Sincerely, 
Elisa Pinat

Reviewer 2 Report

See attachment

Author Response

Dear Reviewer, 

Sincerely, 
Elisa Pinat

Reviewer 3 Report

In this study GNSS reflectometry is used for monitoring long term variations in snow surface height in Antartica on both ice rise and glaciers. On ice an ice rise the seasonal variations in snow surface height was well visible. On moving glaciers, where the data derivation is more complex, the general trend (e.g. meltdown) could be determined.

The article is well written, the employed methods clear and the results supports the conclusions. I suggest accepting it for publications and have just few minor remarks.

Figure 6: the manually measured values could be indicated in the figure.

In discussion a comparison to existing independent data is missing. (e.g. remote sensing )

Author Response

Dear Reviewer, 

Sincerely,
Elisa Pinat

Round 2

Reviewer 1 Report

Most of my concerns were well addressed.